# Proteomic Analysis of a Rice Mutant *sd58* Possessing a Novel *d1* Allele of Heterotrimeric G Protein Alpha Subunit (RGA1) in Salt Stress with a Focus on ROS Scavenging

**DOI:** 10.3390/ijms20010167

**Published:** 2019-01-04

**Authors:** Peng Peng, Yadi Gao, Zhe Li, Yanwen Yu, Hua Qin, Yan Guo, Rongfeng Huang, Juan Wang

**Affiliations:** 1Biotechnology Research Institute, Chinese Academy of Agricultural Sciences, Beijing 100081, China; sxndpp@163.com (P.P.); 15028278721@163.com (Y.G.); lizhe0120@yeah.net (Z.L.); yuyanwen1224@126.com (Y.Y.); qinhua@caas.cn (H.Q.); rfhuang@caas.cn (R.H.); 2State Key Laboratory of Plant Physiology and Biochemistry, College of Biological Sciences, China Agricultural University, Beijing 100193, China; guoyan@cau.edu.cn; 3China National Key Facility of Crop Gene Resources and Genetic Improvement, Beijing 100081, China

**Keywords:** salt tolerance, RGA1, map-based cloning, iTRAQ, ROS scavenging

## Abstract

High salinity severely restrains plant growth and results in decrease of crop yield in agricultural production. Thus, it is of great significance to discover the crucial regulators involved in plant salt resistance. Here, we report a novel mutant, *sd58*, which displays enhanced salt tolerance and dwarf phenotype, by screening from ethyl methane sulfonate (EMS) mutagenized rice mutant library. Genetic analysis showed that *sd58* was caused by a single recessive locus. Map-based cloning and allelic test revealed that the phenotypes of *sd58* were due to the mutation of *RGA1*, encoding the alpha subunit of heterotrimeric G protein (G_α_). A point mutation (G to A) was identified at the splicing site (GT-AG) of the first intron in *RGA1*, which gives rise to the generation of abnormal mRNA splicing forms. Furthermore, 332 differentially abundant proteins (DAPs) were identified by using an Isobaric Tags for Relative and Absolute Quantitation(iTRAQ)-based proteomic technique from seedlings of *sd58* and Kitaake in response to salt treatment. Gene Ontology (GO) and KEGG pathway enrichment analysis revealed these proteins were mainly involved in regulation of the processes such as metabolic pathways, photosynthesis and reactive oxygen species (ROS) homeostasis. Under salt stress, *sd58* displayed lower ROS accumulation than Kitaake, which is consistent with the higher enzyme activities involved in ROS scavenging. Taken together, we propose that RGA1 is one of the regulators in salt response partially through ROS scavenging, which might be helpful in elucidating salt tolerant mechanisms of heterotrimeric G protein in rice.

## 1. Introduction

High salinity is one of the major abiotic stresses in charge of the massive losses in crop yield. This threat becomes more severely because of the growing soil salinization induced by climate changes and irrigation [1,2,3]. Rice, as one of the staple food crops for more than half of the world’s population, is a salt-sensitive crop, with different sensitivity to salt stress at different growth stages [4,5,6,7]. With the aim of improving crop salt tolerance, identifying novel genes and understanding the regulation mechanism in stress adaptation will provide us with the basis for effective engineering strategies.

In general, salinity imposes both ionic and osmotic stresses on plants and causes growth and development impairment [1,2,8]. Plants have evolved multiple physiological and biochemical strategies and responses at both molecular and cellular levels to cope with the stress, such as Salt Overly Sensitive (SOS) pathway, kinases, phosphatases, abscisic acid (ABA), ion transporters, transcription factors [9]. Generally, salinity stress gives rise to oxidative stress on plants through an increase of reactive oxygen species (ROS) contents [9]. On the one hand, ROS functions as an important signal that affects many aspects of plant development such as the cell cycle, programmed cell death as well as hormone signaling [3,4,5,6] and disruption in ROS signaling leads to defects in these developmental processes [7]. ROS overproduction in root in response to salinity stress causes salt tolerance of shoot in *Arabidopsis* through protecting shoot from transpiration-dependent delivery of excess Na^+^ [10]. On the other hand, high ROS concentrations damage proteins, lipids, DNA and carbohydrates [11]. Maintaining cellular ROS homeostasis is an important adaptive trait of salt-tolerant plants in the response to salinity stress. Salt-tolerant rice seedlings have a better protection against ROS by regulating the activities of antioxidant enzymes including catalase (CAT), superoxide dismutase (SOD), peroxidase (POD), ascorbate peroxidase (APX), glutathione reductase (GR) and glutathione peroxidase (GPX) under salt stress [12]. Transgenic *Arabidopsis* plants overexpressing *OsAPXa* or *OsAPXb* exhibit increased tolerance to salinity stress [13]. Constitutive expression of *OsGSTU4* (glutathione S–transferase) in *Arabidopsis* improves the tolerance to salinity and oxidative stress [14]. Therefore, further dissecting the key factors involved in ROS homeostasis is important for genetically promoting plant salt tolerance.

Heterotrimeric G protein consists of G_α_, G_β_ and G_γ_ subunits and performs as a signaling coupler between receptor in membrane and effector in cytoplasm by the exchange of GDP to GTP on the G_α_ subunit [13]. Plant G_α_ subunit is first cloned in *Arabidopsis* and later in rice [14,15]. In rice, the G_α_ subunit is encoded by *RGA1* gene and plays important roles in plant development and stress responses [13,15,16,17,18]. As a null mutant of RGA1 in rice, dwarf1 (*d1*) was initially identified as a gibberellin-insensitive mutant [15,17]. In addition, the function of RGA1 on stress response has been previously reported based on microarray analysis [19], especially hypoxia stress [20], salt-induced cellular senescence and division [13], cold stress [21], drought stress [22]. We further identified the effects of RGA1 on ROS homeostasis and photosynthesis under salt stress, demonstrating the multiple functions of RGA1 on plants stress responses. In the present study, we genetically identified a salt-tolerant and dwarf mutant *sd58* from ethyl methane sulfonate (EMS) mutagenized rice mutant library. *sd58* is an allele of RGA1 loss-of-function mutant *d1*, which also displays enhanced salt resistance. Using Isobaric Tags for Relative and Absolute Quantitation (iTRAQ) proteomic analysis, the differentially abundant proteins (DAPs) of *sd58* under salt stress are enriched in metabolic pathways, photosynthesis and redox homeostasis. Further physiological and biochemical analyses indicated that RGA1 regulates activities of antioxidative enzymes and ROS accumulation under salt stress.

## 2. Results

### 2.1. sd58 Displays Salt Tolerant and Dwarf Phenotypes

To identify key factors involved in salt response, about 10,000 M_2_ population seeds generated by EMS-mutagenesis in Kitaake background were subjected to salt treatment during seedling stage. Among these, a mutant displayed dwarf was named as *sd58* (*salinity-tolerant and dwarf 58*). By self-cross for 3 times, a line of *sd58* exhibited stable phenotype with no segregation was used for further research.

21-d-old seedlings of *sd58* displayed obviously reduced height and dark green leaf, compared with Kitaake. After the treatment of NaCl for 7 d, most leaves of Kitaake turned wilted while those of *sd58* still grew well (Figure 1A,B); and the survival rate of *sd58* mutants after water recovery was 70%, which is 2.5 folds of Kitaake (Figure 1C). Previous reports reveal that salinity stress greatly retards plant growth [23]. The growth of Kitaake and *sd58* seedlings was observed after exposure to high concentrations of NaCl for further insight into RGA1 function (Figure 1D). Statistical analysis indicated that the shoot length of *sd58* is 60% of Kitaake under control condition. The shoot length of Kitaake seedlings decreased by 25% under 100 mmol/L NaCl treatment, whereas no significant decrease was observed in *sd58* seedlings. Additionally, the shoot lengths in both the Kitaake and *sd58* seedlings were significantly restrained with no significant difference under 150 mmol/L NaCl treatment (Figure 1E). Meanwhile, primary root length in both the Kitaake and *sd58* seedlings did not show apparent difference under control condition and 100 mmol/L NaCl treatment. However, the primary root length reduced 90% and 65% in Kitaake and *sd58* seedlings, respectively, under 150 mmol/L NaCl treatment (Figure 1F). Interestingly, although there is no significant difference in the number of crown root development between the Kitaake and *sd58* seedlings before and after salt stress (Appendix A), 5-d-old seedlings of the *sd58* mutant showed increased crown root length compared with Kitaake under both control and salinity conditions (Appendix A). These results indicated that *sd58* alleviates the effects of salinity stress on shoot and root growth.

### 2.2. sd58 is an Allele of d1, with Mutation in Heterotrimeric G Protein Alpha Subunit Encoding Gene RGA1

The genetic analysis was conducted through crossing of *sd58* with Kitaake, using the dwarf phenotype as marker, because it is difficult to evaluate the salinity phenotype in F_2_ individuals. Our result showed that the F_1_ hybrids displayed normal plant height as Kitaake and there were 152 normal and 45 dwarf plants in the F_2_ populations. This segregation of the wild type and the mutant plants fitted a 3:1 segregation ratio (χ^2^ = 0.38 < χ^2^_0.05_, 1 = 3.84), demonstrating that the mutant phenotype in *sd58* is controlled by a single recessive gene. For the map-based cloning, *sd58* was crossed with *indica* variety Dular. 20 and 318 segregated mutant individuals from F_2_ population were used for primary and fine mapping, respectively. Eventually, the mutation locus in *sd58* was mapped to chromosome 5 between insertion-deletion (InDel) markers RM412 and RM422 and the gene was narrowed to a 48 kb interval (Appendix A). According to the available sequence annotation databases (http://rapdb.dna.affrc.go.jp; http://rice.plantbiology.msu.edu), the interval harbors seven open reading frames (ORFs) (Appendix A). Especially, there is a gene *RGA1* (LOC_Os05g26890) that encodes G_α_ subunit of heterotrimeric G protein.

It has been reported that the loss-of-function *d1* mutant of *RGA1* displays shortened internode and small grains [15,16,17]. The *sd58* mutant showed the same phenotypes as *d1*. Thus, *RGA1* was considered as a candidate gene for the *sd58* mutant. By sequencing this locus in *sd58* and Kitaake, single base substitution (G-A) was found at the first base of the first intron in *RGA1* (Appendix A). Moreover, F_1_ hybrids from crosses between *sd58* and *d1* in Nipponbare background showed dwarf phenotype, indicating that *sd58* is a novel allelic mutant to *d1* (Figure 2A). To further confirm the function of RGA1 on salt tolerance, we surveyed the salt sensibility of two reported *d1* mutants in *Nipponbare* and *Shiokari* backgrounds, respectively [15,16]. The results showed that just like *sd58*, both *d1* (Nip) and *d1* (shiokari) exhibited enhanced salt resistance during the seedling stage (Appendix A). Thus, RGA1 plays a negative role in rice salt tolerance.

Sequence analysis showed that the single base substitution in *sd58* located in the splice site of the first intron, leading to defective transcripts of *RGA1*. We synthesized the first strand cDNAs from the total RNAs of *sd58* and Kitaake, respectively and analysis the sequence by fusing the PCR production into T-vector. At least three abnormal splicing forms were found in the *sd58* mutant and the changed splicing forms also obeyed the GT-AG rules (Figure 2B). In these abnormal transcripts, partly or whole of the first intron was retained and encode additional amino acids containing a terminator or frameshift mutation (Figure 2B). Thus, the point mutation leads to abnormal mRNA of *RGA1*, which fail to be translated into functional G_α_ subunit, revealing that *sd58* is a loss-of-function mutation. Taken together, mutation of *RGA1* confers an alleviated sensitivity to salt stress.

### 2.3. Loss-of-Function RGA1 Contributes to Significant Change at Proteome Level under Salinity Stress

To further explore the mechanism underlying the increased salinity tolerance of *sd58* at proteome level, both 14-d-old *sd58* mutant and the Kitaake seedlings were subjected to control or salt stress for 12 h, total proteins from leaves and roots were extracted and analyzed by multiplex run iTRAQ-based quantitative proteomic analysis and LC-MS/MS methods, resulting in the identification of 6421 proteins at a false discovery rate of 1%. Among these proteins, there are 358 up-regulated and 251 down-regulated proteins in the Kitaake seedlings under the short-term salt stress, while only 67 up-regulated and 231 down-regulated proteins were identified in *sd58* mutants (Figure 3A; Appendix A), based on *p*-value < 0.01 and a ≥ 1.2-fold (increased) or ≤ 0.83-fold (decreased) expression change ratio under salt stress relative to the respective controls (Appendix A). Analysis of differentially abundant proteins (DAPs) was a key approach to identify genes that may be responsible for salinity tolerance in the *sd58* mutant. Only a small proportion of DAPs (108) in the Kitaake-salt versus Kitaake-control overlap with the *sd58*-salt versus *sd58*-control DAPs (Figure 3B). However, a large proportion of DAPs in the Kitaake-salt versus Kitaake-control (501) and the *sd58*-salt versus *sd58*-control (190) are unique. Since the *sd58* mutant is more tolerant to salinity than Kitaake, we chose the subset dataset of the DAPs in the *sd58*-salt versus Kitaake-salt comparison excluding DAPs caused by salt effect on Kitaake (in total 332 DAPs), as the subset of core DAPs. These DAPs are more likely to correlate with the salt tolerance of *sd58*, which are used for the following analysis, including 174 up-regulated and 158 down-regulated proteins (Figure 3C, Appendix A).

To gain insight into the functional categories of the DAPs between *sd58* and Kitaake under the treatment with 150 mmol/L NaCl, the 332 core DAPs were categorized according to biological process, cellular component and molecular function using Blast2GO program (http://www.blast2go.org), respectively. The main biological functional categories represented were metabolic process (189), cellular metabolic process (183), biological regulation (50), regulation of biological process (44) and response to stimulus (40). For cellular components, the core DAPs were predominantly distributed in the cell (173), cell part (173), organelle (127), membrane (121), membrane part (105) and organelle part (83). According to the molecular function properties, these proteins were mainly classified into catalytic activity (146) and binding (142) (Figure 4A, Appendix A). These results indicated that many of the DAPs are involved in a wide variety of biological processes including development, metabolism and signal transduction. KEGG pathway enrichment analysis showed that the core DAPs are related to 53 pathways (Appendix A). The top thirty enriched pathways of the core DAPs based on the lowest p values were analyzed. The data showed that the core DAPs are mainly related to metabolic pathways, photosynthesis, ribosome, biosynthesis of secondary metabolites (Figure 4B), suggesting the importance of these processes for salinity resistance in the *sd58* mutant. We focused on the proteomic changes involving the photosynthesis process to address the potential targets modulated by RGA1 under salt stress, which could give rise to the salt tolerance of *sd58* mutant.

### 2.4. sd58 Promotes the Chlorophyll Accumulation and Photosynthesis Rate under Salinity Stress

Comparing proteomes of *sd58* and Kitaake under salt stress revealed a major difference in the expression of photosynthesis-related proteins, photosystem I reaction center subunit VI (Q0DG05, Q6Z3V7)), photosystem II (Q8H4P7, Q6ZBV1), oxygen-evolving enhancer (Q943W1), ATP synthase subunit (P0C2Y7, P0C301, P0C2Z6), ferredoxin related reductase (Q40684, Q6ZFJ3, P41344). Besides, the chlorophyll metabolism related proteins (Q7XKF3) and chlorophyll binding proteins (Q10HD0, Q5ZA98, Q6ZL95, Q6YWJ7) (Table 1, Appendix A). These data indicated that up-regulated photosynthesis related proteins protect rice seedlings from salt-stress impacts by supplying the energy needed to repair salt-induced damage.

Besides the stress sensitivity, comparisons of agronomic traits between *sd58* mutant and Kitaake were surveyed (Appendix A). Consistent with the phenotype at seedling stage, *sd58* showed dramatically reduced plant height at mature stage (Appendix A). The dwarfism of *sd58* is mainly caused by shortened internodes (Appendix A). The growth retardation in *sd58* is also reflected in the aspects of the shortened panicles and leaves (Appendix A). It is interesting that erect flag leaves were observed in *sd58* and this character is considered to have potential in improving production at high crop density (Appendix A) [24]. In *sd58*, the shortened and compact panicles produced similar number grains as those in Kitaake but a few of these grains were unfilled (Appendix A). The filled *sd58* grains also showed decreased grain length and 1000-grain weight, which would seriously affect the rice yield (Appendix A). Overall, the complicated characteristic phenotypes and salt stress insensitivity of *sd58* mutant implied the correlation between plant growth and stress tolerance.

Based on the significant difference of photosynthesis related protein levels between Kitaake and *sd58* under salt stress, in addition to the biomass difference between Kitaake and *sd58* under control condition, we considered whether the *sd58* modulated photosynthesis processes confers to the energy conservation to respond salt stress. The measurement of chlorophylls showed different content of chlorophyll *b*, not chlorophyll *a*, between Kitaake and *sd58* plants. The content of chlorophyll *b* in *sd58* was nearly two times as much as that in Kitaake under both control and salt stress conditions (Figure 5A,B). Accordingly, leaves of *sd58* were wider and greener than that of Kitaake (Appendix A), indicating that *sd58* performs effects on the chlorophyll accumulation and chlorophyll *a*/*b* ratio. Moreover, we measured various parameters determining photosynthetic rate in both *sd58* and Kitaake to find out how photosynthesis is affected by RGA1 in response to salinity. The results showed enhanced photosynthesis in *sd58* mutants, including stomatal conductance, intercellular carbon dioxide concentration, net photosynthetic rate and transpiration rate under both control and salt stress conditions (Figure 5C–F), suggesting that higher photosynthesis capacity in *sd58* mutant. These results indicated that the regulation of *sd58* on photosynthesis via accumulated chlorophyll, implying that enhanced photosynthesis capacity confers to the salt tolerance.

### 2.5. sd58 Mutant Confers the Tolerance to Salinity Tolerance Partially through Enhancing Antioxidation Capacity

Plants suffering from abiotic stress often exhibit symptoms of oxidative stress as evidenced by enhanced accumulation of malondialdehyde (MDA) and reactive oxygen species (ROS), reflecting the damage extent of lipid peroxidation under salt stress [4]. No significant differences in the contents of MDA equivalents between Kitaake and *sd58* seedlings were found under control condition (Figure 6A). Significant increases of MDA contents in the Kitaake seedlings were observed after exposure to salt stress, while the sd58 mutant plants maintained a relatively constant MDA contents when challenged by the identical salt stress, leading to significantly lower MDA contents in the *sd58* mutant than that in the Kitaake under salt conditions (Figure 6A). Additionally, histochemical analysis was performed to detect ROS accumulation by NBT and DAB staining. Under treatment with 150 mmol/L NaCl for 7 d conditions, increased H_2_O_2_ and O_2_·- levels were detected in both *sd58* and Kitaake, while the levels of H_2_O_2_ and O_2_·-—in *sd58* were significantly lower than that in Kitaake (Figure 6B–D). These results suggest that mutation of rice G_α_ subunit exhibits significantly less ROS accumulation during seedling growth under salinity stress.

It has been reported that various well-developed antioxidative mechanisms to regulate the ROS level, including antioxidant enzymes such as superoxide dismutase (SOD), catalase (CAT) and peroxidase (POD), ascorbate peroxidase (APX) as well as glutathione reductase (GR) and glutathione peroxidases (GPX) [11]. Based on the improved protein levels of APX and GPX in proteomic profile of *sd58* under salt stress (Table 1), we furtherly detected the activities of ROS-scavenging enzymes after exposed to NaCl treatment. Under control condition, activities of CAT, POD, GR and GPX were higher in shoots of *sd58* than those in Kitaake (Figure 7), with except for the comparable activities of SOD and APX in both *sd58* and Kitaake (Figure 7C, D). Accordingly, there were marked increases in activities of these enzymes for plants upon exposure to salt stress and significantly higher enzymatic activities in *sd58* were observed than those in Kitaake under salt stress (Figure 7). Together, our findings suggest that the *sd58* mutant is equipped with greater ability to eliminate ROS and more tolerance to the oxidative stress associated with high salinity conditions.

## 3. Discussion

High salinity may adversely affect rice growth and development by impairing various physiological and biochemical processes, including membrane permeability, ion balance, photosynthesis, stomatal aperture, redox homeostasis and ROS production and detoxification [25]. Therefore, the attempts to understand the mechanism of salt response might be an effective means to improve the crop salt tolerance. Here, a salt-tolerant and dwarf mutant *sd58* from the EMS mutant library, a novel allele of *d1*, was able to alleviate the salt stress effect on shoot and root growth at seedling stage, indicating that RGA1 plays a negative role in salt resistance. Furthermore, iTRAQ-based proteomic analysis of *sd58* not only indicated that multiple pathways, including photosynthesis, carbohydrate and energy metabolism are related to plant growth under salinity stress but also evidenced the potential role of RGA1 in ROS scavenging under salt stress conditions. Especially, antioxidant machinery of *sd58* was greatly enhanced under salt stress conditions, which is consistent with the lower ROS accumulations.

Several alleles of *d1* have been reported, including various alleles of *d1* [16], *T65d1* with two bases deletion [17] and *HO541* with a large-fragment deletion [15], which results in non-functional G_α_ subunit. Another allelic mutant *Epi-d1* represents a metastable epigenetic mutant correlated with repressive histone and DNA methylation marks in the D1 promoter region [26]. Here, we revealed that *sd58* displayed similar phenotype as the reported loss-of-function mutant of RGA1, such as dwarfism, small grains, compact panicles, abnormal leaf blade morphology and dark-green leaves. Furthermore, *sd58* had mutated at the GT-AG splicing site of the first intron in RGA1 with a substitution (G-A) and the mutation of splicing site in *sd58* resulted in abnormal transcripts of RGA1 that cannot produce functional G_α_ subunit. Thus, the *sd58* is a novel allele of *d1* with different transcripts of *RGA1*.

Proteomics analysis of salt-tolerant and salt-sensitive plants could imply some important molecular-biochemical markers mediated salt response. In our iTRAQ data, most of different expressed proteins between *sd58* and Kitaake are correlated with photosynthesis, which is also demonstrated in the previous report that RGA1 plays a role in photoprotection and photo avoidance in rice [18]. Salinity stress can decrease chlorophyll contents and suppress photosynthesis in various crop cereals [4]. Our results also showed that *sd58* retained more chlorophyll content and higher photosynthetic rates than Kitaake under salinity stress. When plants are under salt stress, stomatal closure and the corresponding limited CO_2_ uptake lead to inadequate photosynthesis [27]. Decreased photosynthetic rate in salt-stressed leaves was mainly attributed to low CO_2_ concentrations in the chloroplasts, which was determined by stomatal and mesophyll conductance [28]. Here, *sd58* exhibits increased stomatal conductance, intercellular CO_2_ concentration than Kitaake, indicating that stomatal limitations significantly affected the photosynthetic performance of the rice leaves, which is consistent basically with the drought resistance of *d1* [22]. Thus, the proteomics analysis of *sd58* in this study supplies evidences for integrated profile of RGA1 in salt response.

In general, the common damage anticipated under stress conditions is the accumulation of excessive ROS, as byproducts of photosynthesis, respiration, photorespiration. Increased production of ROS is a common consequence under various abiotic stresses including salinity [29,30,31,32]. Meanwhile, it is well known that ROS are generated in abundance by photosynthesis [33]. Although higher photosynthetic rates of *sd58* mutant play potential role in enhanced salt tolerance via supplying energy and carbon source for plant growth under salinity stress, the ROS scavenging is also improved in *sd58*. Our results demonstrated that the down regulation of O_2_·- and H_2_O_2_- accumulations in *sd58*, together with the maintenance of a lower MDA compared with those observed in Kitaake, suggested that the *sd58* mutant exhibited greater tolerance to oxidative stress under salinity stress. To avoid excessive accumulation of ROS under salt stress, plants possess efficient antioxidant system including enzymatic and non-enzymatic which can protect the plant cells from ROS induced oxidative damage [9]. Here, the antioxidant enzymes such as SOD, CAT, POD, APX, GR and GPX showed significantly higher activity under salinity stress in *sd58* than those in Kitaake. These results indicated that *sd58* maintained lower ROS levels through up-regulating activities of antioxidant enzymes, thus protecting the photosynthetic apparatus and preserving the plants growth under salinity stress. However, the molecular mechanism of RGA1 modulation on ROS homeostasis needs further research.

Taken together, this study revealed the mechanism of the negative regulatory function of RGA1 in initial response to salinity in rice and proposed a hypothetical model (Figure 8). Based on the proteomic analysis of *sd58* under salt stress, RGA1 affects photosynthesis and chlorophyll metabolism related proteins levels, which was partly mediated by the regulations on stomatal conductance. Meanwhile, RGA1 acts upon redox balance through affecting antioxidant enzymes activities and ROS homeostasis, further impacting on photosynthesis and stomatal conductance. Therefore, RGA1 could be an ideal candidate to be modified for developing a variety of crop tolerant to salinity stress.

## 4. Materials and Methods

### 4.1. Plant Materials and Growth Conditions

Four rice cultivars (Oryza sativa L. cv. Kitaake, Nipponbare, Shiokari and Dular) and two alleles of *d1* mutant in Nipponbare and Shiokari backgrounds respectively were used in this study. Rice seedlings of all materials were grown in the growth chamber at 60% relative humidity with 16 h white light (200 µmol/m^2^/s)/8 h dark at 30 °C/25 °C (day/night). For rice cross, propagation and observation of agronomic traits, plants were grown in the field under natural conditions.

For salinity stress in soil, germinated seeds were grown in pots containing filled mixture of soil and vermiculite (1:1). 21-d-old plants were irrigated with 150 mmol/L NaCl solution every day, followed with wilting rate statistics after 7 days. The control plants were irrigated with water every day. For water recovery, the treated plants were irrigated with water every day to remove salt stress, followed with survival rate statistics after 14 days.

### 4.2. Screening of Salt-Tolerant Mutants

For ethyl methane sulfonate (EMS) mutagenesis, rice (Oryza sativa L. cv. Kitaake) seeds were soaked in water for 12 h and then treated with 0.5 mol/L EMS (Sigma-Aldrich Corp., St. Louis, MO, USA) for 8 h. M_2_ population seeds were used for the screen of salt-tolerant mutants. After germination at 30 °C for 2 d, the seeds were sown into soil in basin. And 21-d-old seedlings were watered with 150 mmol/L NaCl for 5 d and the seedlings exhibited salt-tolerant phenotypes with greener and less wilted leaves were transplanted to soil in the field for harvesting seeds.

### 4.3. Measurement of the Root Growth under Salinity Stress

For the root elongation of rice seedlings assays under salinity stress, germinated seeds were placed on stainless nets in water with or without NaCl. The seedlings were grown at 28 °C for 3 d. The seedling roots were photographed and the root length was measured by digitized images using Image J software.

### 4.4. Genetic Analysis and Map-Based Cloning

The *sd58* mutant was crossed with wild type Kitaake and the plant height of F_2_ population was evaluated to determine the loci number and dominant/recessive of the mutation. F_2_ mapping populations were generated from a cross between *sd58* mutant and *indica* cultivar Dular. Genomic DNA was isolated from seedlings with mutant phenotype. Primary mapping was performed by bulked segregate analysis with 20 mutant individuals, using 123 SSR markers as previously described [34]. Then 318 mutant individuals selected from F_2_ population were used for fine mapping. The mutation site was mapped to chromosome 5 between a 59.02 kb region and the candidate gene was confirmed by DNA sequencing and allelic test through the cross between *sd58* and reported allele. The markers and primers used for the map-based cloning were listed in Appendix A.

Genomic DNA and total RNAs were extracted from Kitaake and *sd58*, respectively. The complementary DNAs were synthesized using the cDNA synthesis kit (Vezyme, Nanjing, China). The genomic polymerase chain reaction (PCR) and reverse transcription (RT)-PCR were performed with specific primers (Appendix A). The PCR products were sequenced directly or after fused into T-vector by the Lethal Based Fast Cloning Kit (Tiangen, Beijing, China).

### 4.5. Protein Extraction, Protein Digestion and iTRAQ Labelling

Total proteins were extracted from the roots and leaves of 21-d-old rice seedlings according to the following procedure. The tissues were immediately immersed in liquid nitrogen and resuspended in the Lysis buffer (6 M Urea, 4% CHAPS, *w*/*v*, 2 M Thiourea, 40 mmol/L Tris-HCl, pH 8.5, 1 mmol/L PMSF, 2 mmol/L EDTA). The proteins were reduced by dithiothreitol (10 mmol/L, final concentration) at 56 °C for 1 h and then alkylated by 55 mmol/L iodoacetamide (final concentration) in the darkroom for 1 h. The reduced and alkylated protein mixtures were precipitated by adding 4 volumes of chilled acetone at −20 °C overnight.

The dissolved protein was analyzed for protein concentration using the Protein Assay Kit (Bio-Rad, Hercules, CA, USA) based on the Bradford method using a BSA standard. The quality of each protein sample was evaluated by SDS-PAGE. Samples showing no degradation and good resolution with low background were kept at −80 °C for further analysis. 100 μg total proteins were digested (36 °C, 16 h) with Gold grade Trypsin (Promega, Madison, WI, USA) with the ratio of 30:1 (protein: trypsin). Then the peptides were dried by vacuum centrifugation and reconstituted in 0.5 M TEAB and labeled using iTRAQ 4-plex kits according to the manufacturer’s manual (AB Sciex Inc., Framingham, MA, USA). For labeling, the control biological replicate samples, Kitaake-control and *sd58*-control were labeled with iTRAQ tags 116 and 118 and the salt-treated biological replicate samples Kitaake-salt (salt stress) and *sd58*-salt (salt stress), labeled with tags 119 and 121, respectively. Two biological replicates were carried out for each sample.

### 4.6. Strong Cation Exchange (SCX) Fractionation and LC-ESI-MS/MS Analysis by Q Exactive

iTRAQ labeled peptides were fractionated by SCX chromatography using the AKTA Purifier system (GE Healthcare, Chicago, IL, USA). The dried peptide mixture was reconstituted and acidified with 2 mL buffer A (10 mmol/L KH_2_PO_4_ in 25% can, *v*/*v*, pH 2.6) and loaded onto a PolySULFOETHYL 4.6 × 100 mmol/L column (5 µm, 200 Å, PolyLC Inc, Columbia, MD, USA). The peptides were eluted at a flow rate of 1 mL/min with a gradient of 0–10% buffer B (500 mmol/L KCl, 10 mmol/L KH_2_PO_4_ in 25% ACN, pH 2.6) for 2 min, 10–20% buffer B for 25 min, 20–45% buffer B for 5 min and 50–100% buffer B for 5 min. The elution was monitored by absorbance at 214 nm and fractions were collected every 1 min. The eluted peptides were pooled as 12 fractions, desalted using a Strata X C18 column (Phenomenex) and vacuum dried. All samples were stored at −80 °C until LC-MS/MS analysis.

Eight fractions were collected and used for LC-MS/MS analysis. Experiments were performed on a Q Exactive mass spectrometer that was coupled to an EasynLC (Proxeon Biosystems, now Thermo Fisher Scientific, Waltham, MA, USA). 10 mL of each fraction were injected for nano LC-MS/MS analysis. The peptide mixture (5 µg) was loaded onto a C18 reversed-phase column (Thermo Scientific Easy Column, 10 cm long, 65 µm inner diameter, 3 µm resin) in buffer A (0.1% formic acid, *v*/*v*) and separated with a linear gradient of buffer B (80% acetonitrile and 0.1% formic acid, *v*/*v*) at a flow rate of 250 nL/min controlled by IntelliFlow technology over 140 min. MS data were acquired using a data-dependent top10 method dynamically choosing the most abundant precursor ions from the survey scan (300–1800 *m*/*z*) for higher-energy collisional dissociation fragmentation. Determination of the target value was based on predictive automatic gain control. Dynamic exclusion duration was 60 s. Survey scans were acquired at a resolution of 60,000 at *m*/*z* 200 and the resolution for higher-energy collisional dissociation spectra was set to 16,500 at *m*/*z* 200. Normalized collision energy was 30% and the underfill ratio, which specifies the minimum percentage of the target value likely to be reached at maximum fill time, was defined as 0.1%. The instrument was run with peptide recognition mode enabled.

### 4.7. Proteomic Bioinformatic Analysis

The MS and MS/MS data were searched against the UniProt rice database (Uniprot_oryza_sativa_147047_20151117.fasta, download time: 17 November 2015) using Mascot 2.3.02. The peptide and protein data were extracted using high peptide confidence and top one peptide rank filters. High confidence peptide identifications were obtained by setting a target FDR threshold of 1% at the peptide level. Relative quantitation of proteins was performed based on the relative intensities of reporter ions released during the MS/MS peptide fragmentation. For each replicate of proteomics, iTRAQ ratios between salt stressed plants and controls for each run were converted to z-scores to normalize the data.

The annotation of the identified proteins was carried out based on the molecular functions, cellular components and biological processes listed in the Gene Ontology (GO) database (http://www.geneontology.org) and blast2go (http://www.blast2go.com/) in compliance with GO standards. Pathway grouping was performed using the software KOBAS 3.0, which stands for Kyoto Encyclopedia of Genes and Genomes (KEGG) Orthology-based Annotation System (http://kobas.cbi.pku.edu.cn). GO and pathway enrichment analysis were performed to determine which functional subcategories and metabolic pathways were over represented by the differentially accumulated proteins. Pathways with FDR-corrected *p*-values < 0.05 were considered statistically significant and were displayed by color intensity.

### 4.8. Quantitative Analysis of Chlorophyll Content, Photosynthesis Rate and Chlorophyll Fluorescence

Chlorophylls were extracted from 0.1 g leaves of 21-d-old seedlings and the contents of chlorophyll *a* and *b* were performed as described previously [35]. To determine photosynthesis parameters, the plants were planted in greenhouse for about 20 d before measurement. Photosynthesis (P) and transpiration (T) rates were measured using a portable photosynthesis system (LI-6400XT) in the morning (9 to 11 a.m.). All the photosynthetic measurements were taken at a constant air flow rate of 500 μmols^−1^. The concentration of CO_2_ was 400 μmol using the system’s CO_2_ injector and the temperature was maintained at 30 °C and the photosynthetic pho-ton flux density was 800 μmol (photon) m^−2^ s^−1^. Three measurements were made for each plant and 5 plants were used for replicates.

### 4.9. Visualization of Reactive Oxygen Species

The hydrogen peroxide and superoxide anion radicals were visualized by 3,3′-diaminobenzidine (DAB) and nitroblue tetrazolium (NBT) staining, respectively [36,37]. In brief, rice leaves from triplicate biological replicates of the samples were infiltrated in 0.1% DAB (*w*/*v*) or 0.1% NBT (*w*/*v*). Infiltration was carried out by building up a vacuum (∼10–15 kPa) and until the leaves were completely infiltrated. The incubation was conducted in a growth chamber in the dark overnight.

### 4.10. Determination of Malondialdehyde Equivalents

Malondialdehyde (MDA) equivalents content in rice leaves was determined following the protocols described by [38]. Briefly, rice leaves were weighted and homogenized in 5 mL 10% trichloroacetic acid (TCA) (*w*/*v*) solution and then centrifuged at 10,000× *g* for 10 min. Thereafter, 2 mL supernatant was added in 2 mL 10% (*w*/*v*) TCA containing 0.6% thiobarbituric acid (TBA) (*w*/*v*). The mixture was then incubated in water at 95 °C for 30 min and the reaction was stopped in an ice bath. The absorbance of the solution was measured at 450, 532 and 600 nm, respectively. Five biological replicates of each treatment were used.

### 4.11. Determination of Peroxidase, Superoxide Dismutase and Catalase Activity

0.5 g fresh weight of leaves were grounded thoroughly with a cold mortar and pestle in 50 mmol/L potassium phosphate buffer (pH 6.8) containing 1% polyvinylpyrrolidone (*w*/*v*). The homogenate was centrifuged at 15,000× *g* for 20 min at 4 °C. The supernatant was crude enzyme extraction. Fresh leaf samples were used for enzyme extraction. All operations were carried out at 4 °C. The activities of peroxidase (POD; EC 1.11. 1.7), superoxide dismutase (SOD; EC 1.15.1.1), catalase (CAT; EC 1.11. 1.6), ascorbate peroxidase (APX; EC 1.11.1.11), glutathione reductase (GR; EC 1.8.1.7) and glutathione peroxidases (GPX; EC 1.11.1.9) were measured using the protocols described before [39].

### 4.12. RNA Isolation and Synthesis of First-Strand cDNAs

21-d-old rice seedlings were used for the extraction of RNA by using TRIZOL reagent (Invitrogen, Carlsbad, CA, USA) and then reverse transcribed to the first-strand cDNAs with M-MLV reverse transcriptase (Vezyme).

## Figures and Tables

**Figure 1 ijms-20-00167-f001:**
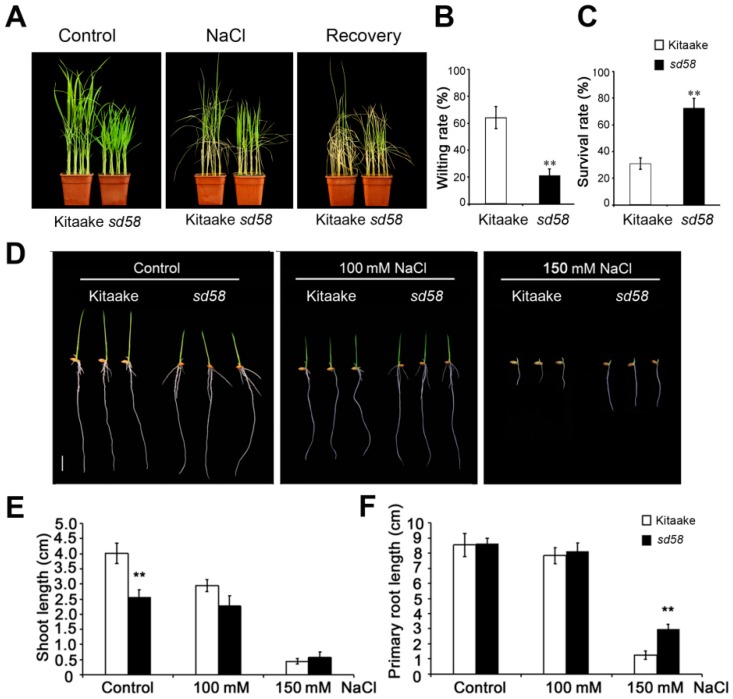
The *sd58* mutant displays salt tolerance phenotype. (**A**)The phenotype of *sd58* mutant under salt stress. Wilting rates (**B**) and survival rates (**C**) of the rice seedlings in (**A**) after the treatment of 150 mmol/L NaCl and water recovery, respectively. The phenotypes of 3-d-old seedlings under salt stress (**D**) and statistics data including shoot length (**E**), primary root length (**F**) of *sd58* and Kitaake grown under 0, 100 mmol/L, 150 mmol/L NaCl after germination. Bars =1 cm. Values are means ± SD. Student’s *t*-tests were used to assess the significant differences between *sd58* and Kitaake under the same conditions (** *p* < 0.01).

**Figure 2 ijms-20-00167-f002:**
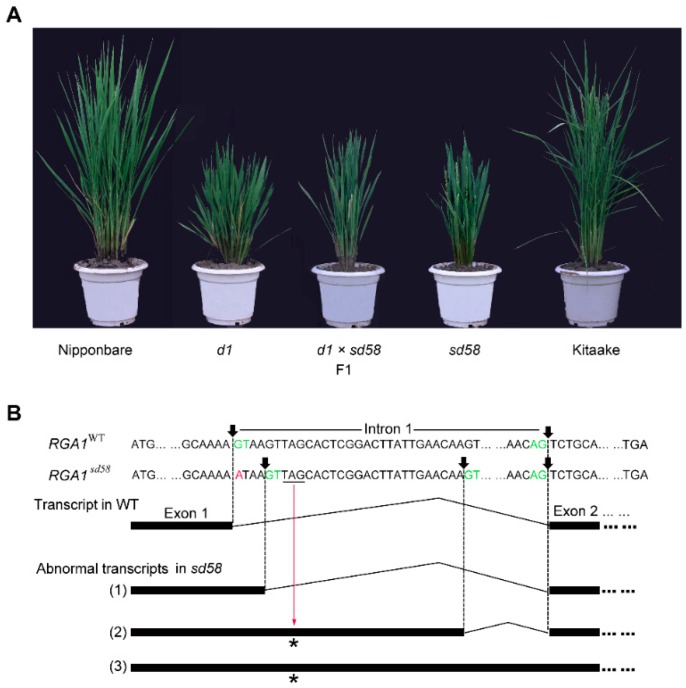
*sd58* is an allele mutant of *RGA1*. (**A**) Allelic test of *sd58* mutant through crossing with *d1*, an allelic mutant of *RGA1* in the Nipponbare background. (**B**) Three types of abnormal transcripts were identified in *sd58*. Transcript (1) caused frame shift and transcript (2) and (3) generated a stop codon in advance. Arrows indicate the splicing site and GT-AG sites are shown in green. Mutated nucleotide is shown in red and the stop codons induced by mutation are underlined and indicated by asterisks.

**Figure 3 ijms-20-00167-f003:**
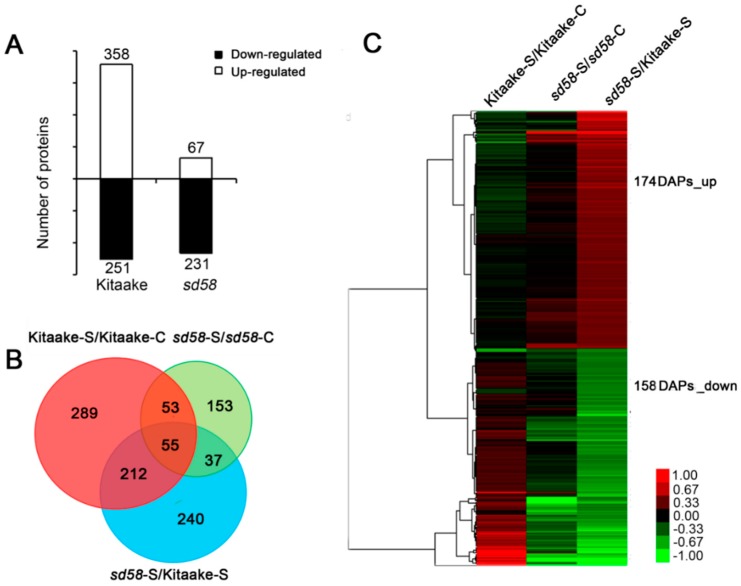
Differentially abundant proteins (DAPs) are identified in Kitaake and *sd58* plants under salinity stress. (**A**) The numbers of up- and down-regulated (DAPs in Kitaake and *sd58* under salinity stress, respectively. (**B**) Venn diagram displaying the overlaps among different groups of DAPs. The numbers in each circle (Kitaake-salt versus Kitaake-control, *sd58*-salt versus *sd58*-control and *sd58*-salt versus Kitaake-salt) indicate the total number of different DAPs in each comparison group and the numbers in the overlapping areas are the number of shared DAPs between comparison groups. (**C**) Heat map showing the core DAPs with different expression patterns in *sd58* and Kitaake under salinity stress.

**Figure 4 ijms-20-00167-f004:**
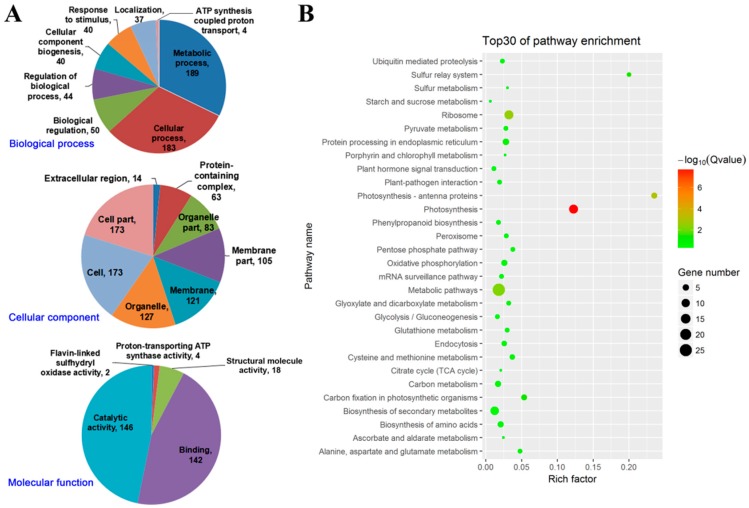
Gene Ontology (GO) enrichment analysis and KEGG pathways enriched of core DAPs. (**A**) Significantly enriched GO terms at biological process (top), cellular component (middle) and molecular function (down) ontology level for core DAPs. GO terms were defined as significant enriched if false discovery rate (FDR) was ≤0.05. (**B**) The top thirty enriched pathways by core DAPs determined based on the lowest over-represented *p* values were analyzed. The number of DAPs in the pathway is indicated by the circle area and the circle color represents the ranges of the corrected *p* value.

**Figure 5 ijms-20-00167-f005:**
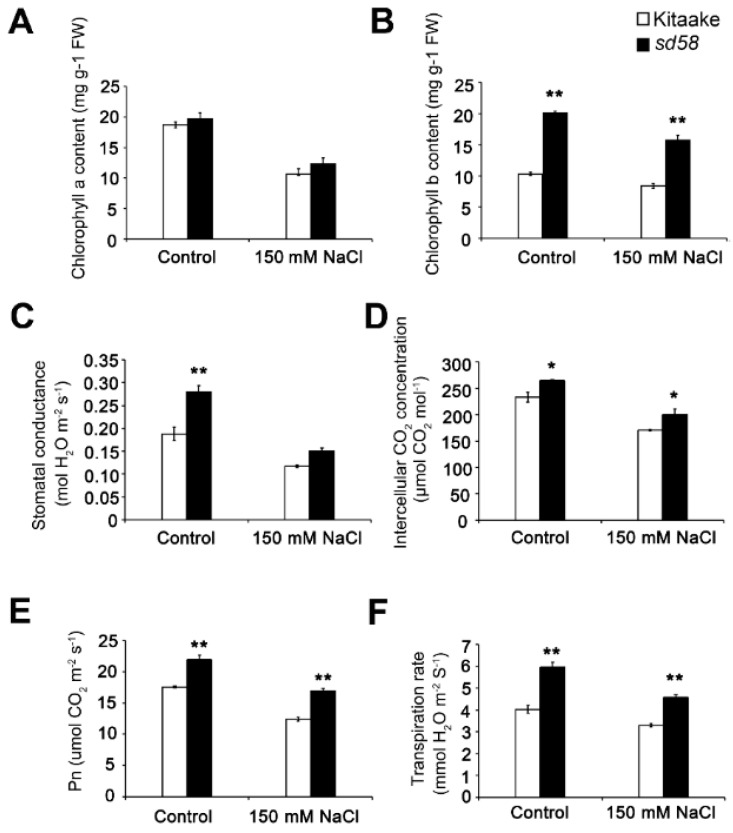
Effect of salt stress on photosynthesis physiological characterization in Kitaake and *sd58* seedlings. Measurement of chlorophyll *a* contents (**A**), chlorophyll *b* contents (**B**), stomatal conductance (**C**), intercellular CO_2_ concentration (**D**), photosynthesis rate (**E**) and transpiration rate (**F**) of Kitaake and *sd58* seedlings under 150 mmol/L NaCl treatment. Data are presented as mean ± SD (n = 3, * *p* < 0.05, ** *p* < 0.01, Student’s *t*-test).

**Figure 6 ijms-20-00167-f006:**
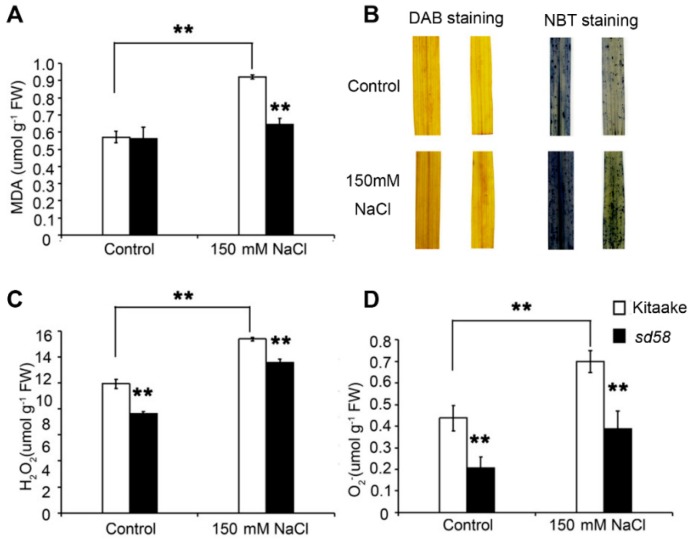
*sd58* exhibits less ROS accumulation under salinity stress. (**A**) Quantitative analysis of malondialdehyde (MDA). (**B**) Hydrogen peroxide and superoxide anion radicals in 2-week-old Kitaake and *sd58* seedlings under 150 mmol/L NaCl treatment for 7d were detected with 3,3′-diaminobenzidine (DAB) and nitroblue tetrazolium (NBT) staining. (**C**,**D**) Quantitative analysis of hydrogen peroxide (H_2_O_2_) and superoxide anion radicals (O_2_·-) in (**B**). Three replica experiments were performed. Data are means ± SD (n ≥ 4, ** *p* < 0.01, Student’s *t*-test).

**Figure 7 ijms-20-00167-f007:**
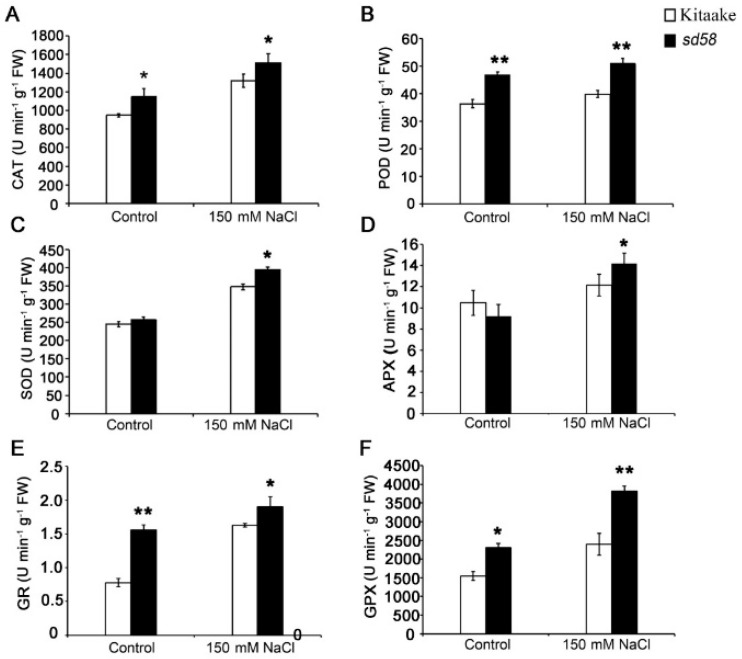
*sd58* exhibits enhanced antioxidation capacity under salinity stress. (**A**–**F**) Detection of, catalase (CAT), peroxidase (POD), superoxide dismutase (SOD), ascorbate peroxidase (APX), glutathione reductase (GR) and glutathione peroxidases (GPX) activities in the leaves of rice seedlings treated with 150 mmol/L NaCl. Data are means ± SD (n ≥ 4, * *p* < 0.05, ** *p* < 0.01, Student’s *t*-test).

**Figure 8 ijms-20-00167-f008:**
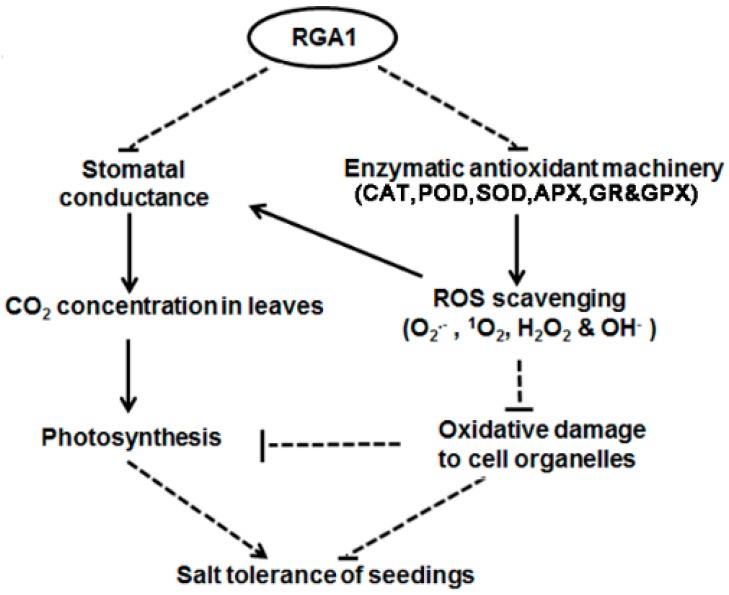
A hypothetical model for RGA1 mediated salinity tolerance through the integration between photosynthesis and ROS homeostasis. RGA1 regulates activities of antioxidant enzymes to affect ROS homeostasis, which confers to oxidative damage on cell organelles. In another hand, RGA1-regulated stomatal closure was associated with photosynthetic efficiency, which could be triggered by ROS. Overall, RGA1 plays complicated roles in response to salt stress through balance between photosynthesis and oxidative damage. Arrows and T-bars indicate activation and inhibition, respectively. Broken lines indicate indirect regulations.

**Table 1 ijms-20-00167-t001:** Categories of proteins that accumulate differently in *sd58* and Kitaake at seedling stage under salt stress.

Categories	Accession	Description	Peptide	Unique Peptide	Coverage	PSMs	*sd58*-C Versus Kitaake-C	Kitaake-S Versus Kitaake-C	*sd58*-S Versus *sd58*-C	*sd58*-S Versus Kitaake-S
Photosynthesis	Q0DG05	Photosystem I reaction center subunit VI	4	4	39.44	61	1.194687663	0.884711293	1.021167747	1.378954377
	Q8H4P7	photosystem II 10 kDa polypeptide	4	4	24.09	29	1.546765379	0.90177992	0.963631996	1.652856287
	Q943W1	oxygen-evolving enhancer protein 1	22	22	64.56	291	1.148000937	0.908757715	1.058357592	1.336985081
	P0C2Y7	ATP synthase subunit a	2	2	3.24	7	1.189606815	0.905690444	1.040516165	1.366697782
	Q40684	Ferredoxin	1	1	6.76	2	5.256444569	1.753843469	0.058994804	0.176813339
	P0C301	ATP synthase subunit c	1	1	8.64	2	1.06387461	0.977608316	1.230407281	1.338981108
	Q7XTG4	Membrane protein	1	1	4.93	6	1.080830998	0.945503375	1.124059544	1.284943482
	P0C2Z6	ATP synthase subunit alpha	26	2	57.59	310	1.396674248	0.838719483	1.10045561	1.832529283
	Q6ZBV1	photosystem II 10 kDa polypeptide	3	3	23.08	48	1.079804792	0.931293627	1.12603067	1.305596084
	Q6ZFJ3	Ferredoxin--NADP reductase	23	19	58.74	149	1.150440812	0.955853923	1.077095101	1.296363527
	P41344	Ferredoxin--NADP reductase,	23	19	58.29	155	1.113047516	0.928829949	1.074319895	1.287392909
	Q6Z3V7	Photosystem I reaction center subunit VI	3	3	42.28	44	1.21137611	0.948516534	1.17008755	1.494350445
Porphyrin and chlorophyll metabolism	Q7XKF3	Protochlorophyllide reductase A	18	18	60.21	46	1.244582113	0.977177448	0.996806607	1.269582792
Photosynthesis- antenna proteins	Q10HD0	Chlorophyll *a*-*b* binding protein	7	4	31.94	186	1.161584754	0.96898888	1.071110401	1.284003911
	Q5ZA98	Chlorophyll *a*-*b* binding protein	6	6	17.01	45	1.210805009	0.993780608	1.053959837	1.284126335
	Q6ZL95	Chlorophyll *a*-*b* binding protein	3	3	21.67	19	1.381846156	0.894720871	0.994724709	1.536296471
	Q6YWJ7	Chlorophyll *a*-*b* binding protein	6	6	35.66	39	1.235309035	0.951992885	1.070666181	1.389299887
Carbon metabolism	Q7X8A1	Glyceraldehyde-3-phosphate dehydrogenase	19	15	53.23	269	1.185175004	0.893344399	1.068761987	1.417896607
	Q9SE42	Ribulose-phosphate 3-epimerase	4	4	25.88	12	1.125852891	0.92458396	1.103994095	1.344318091
	Q7XZW5	Malate dehydrogenase	15	13	52.26	77	1.236084716	0.958486603	1.042411695	1.344316299
	Q40677	Fructose-bisphosphate aldolase	19	18	53.35	220	1.164792195	0.983320615	1.086516982	1.287033427
Peroxisome	Q9FW35	pex14 protein	2	2	5.39	5	1.313339411	0.994865066	1.019323065	1.345626859
	B7EBN1	Os02g0225000 protein	2	1	6.61	7	1.19598591	0.930099311	0.991836807	1.275372245
Glutathione metabolism	Q6ZJJ1	Probable L-ascorbate peroxidase 4	14	12	44.67	69	1.204637136	0.986577028	1.057649443	1.291418471
	B7FAE9	Glutathione peroxidase	8	6	45.3	25	1.189347746	1.034130023	1.128526366	1.297912506

The red and green numbers respectively indicate proteins that are elevated and attenuated, in amount in *sd58* seedlings. S and C respectively indicate salt stress and control condition.

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
