# Peer review of "Proteomic Analysis of a Rice Mutant sd58 Possessing a Novel d1 Allele of Heterotrimeric G Protein Alpha Subunit (RGA1) in Salt Stress with a Focus on ROS Scavenging"

_ijms, 2019, doi:10.3390/ijms20010167_

Round 1

Reviewer 1 Report

In the present manuscript the identification and characterization of an allele of RGA1 is described. In this respect the authors focused on the salt tolerance of this novel mutant.

The strength of this work is that a broad range of methods has been applied to fulfill this task: They range from map based cloning over genetic and proteomic analyses (and their bioinformatic evaluation) to physiological investigations of phtosynthesis and ROS production and scavenging.

In the course of this studies the authors could show that the latter processes are highly dependent on salinity in the mutant and that their increased tolerance may be correlated with their ability to eliminate ROS to a higher extent. Also photosynthetic parameters are affected in the mutants (Fig. 5), but in this case most of them are not specifically reacting to salinity. Most of them are rather different even under control conditions. To conclude that more chlorophyll B and enhanced transpiration rate leads to more salinity tolerance should be discussed more thoroughly. Also the model in Fig 8 is highly speculative in this regard. Possibilities to prove this correlation between photosynthesis and ROS scavenging should be discussed.

I also missed a critical discussion on the fact that the sd58 mutant reveals a drastic reduction in grain weight (Table S6). To announce the mutation of RGA1 on hand side as a way for the breeding of salt tolerant rice plants and this downside on the other hand shoul appear in the discussion and must not be hidden in the results section.

The major flaw of this manuscript is its language and style: There are an extraordinary high number of mistakes in orthography and grammar in this text. They sometimes lead to non-understandable sentences, e.g. for lines 21-23 (in the abstract) or lines 228-230. I have to insist that this text has to be read and edited by a native english speaking scientist before resubmission.

Author Response

Response to Reviewer 1 Comments

Point 1: In the course of this study, the authors could show that the latter processes are highly dependent on salinity in the mutant and that their increased tolerance may be correlated with their ability to eliminate ROS to a higher extent. Also photosynthetic parameters are affected in the mutants (Fig. 5), but in this case most of them are not specifically reacting to salinity. Most of them are rather different even under control conditions. To conclude that more chlorophyll B and enhanced transpiration rate leads to more salinity tolerance should be discussed more thoroughly. Also the model in Fig 8 is highly speculative in this regard. Possibilities to prove this correlation between photosynthesis and ROS scavenging should be discussed.

Response 1: Thanks for the constructive suggestion. Although the photosynthetic parameters in sd58 mutant are not specifically increased under salt stress, higher photosynthetic rates of sd58 mutant play potential role in enhanced salt tolerance via supplying energy and carbon source for plant growth under salinity stress. It is well known that ROS are generated in abundance by photosynthesis and chloroplast has mechanisms to photoprotection (Foyer, 2018). We have added “Meanwhile, it is well known that ROS are generated in abundance by photosynthesis [33]. Although higher photosynthetic rates of sd58 mutant play potential role in enhanced salt tolerance via supplying energy and carbon source for plant growth under salinity stress, the ROS scavenging is also improved in sd58 in the discussion of the revised manuscript.

Point 2: I also missed a critical discussion on the fact that the sd58 mutant reveals a drastic reduction in grain weight (Table S6). To announce the mutation of RGA1 on hand side as a way for the breeding of salt tolerant rice plants and this downside on the other hand should appear in the discussion and must not be hidden in the results section.

Response 2: Thanks very much for the criticisms. The small grain phenotype of sd58 has been characterized in the results section 2.4 in the submitted manuscript. The mutation of RGA1 indeed confers to the decreased yield, which is correlated with the function of RGA1 in plant growth depending on phytohormone signalling. However, the enhanced salt tolerance of mutants is useful for planting in salinity land and crop genetic improvement.

Point 3: The major flaw of this manuscript is its language and style: There are an extraordinary high number of mistakes in orthography and grammar in this text. They sometimes lead to non-understandable sentences, e.g. for lines 21-23 (in the abstract) or lines 228-230. I have to insist that this text has to be read and edited by a native English speaking scientist before resubmission.

Response 3: Thanks for reminding us. We have asked a native English speaking colleague to read and edit our manuscript.

We have corrected the sentence in lines 21-23 as “Map-based cloning and allelic test revealed that the phenotypes of sd58 were due to the mutation of RGA1, encoding the alpha subunit of heterotrimeric G protein (Gα). A point mutation (G to A) was identified at the splicing site (GT-AG) of the first intron in RGA1, which gives rise to the generation of abnormal mRNA splicing forms”.

We have corrected the sentence in lines 228-230 as “Based on the significant difference of photosynthesis related protein levels between Kitaake and sd58 under salt stress, in addition to the biomass difference between Kitaake and sd58 under control condition, we considered whether the sd58 modulated photosynthesis processes confers to the energy conservation to respond salt stress”.

Reviewer 2 Report

Dear Sir or Madam,

the manuscript „ Proteomic analysis of a novel d1 allele reveals the regulation of rice heterotrimeric G protein alpha subunit (RGA1) in salt stress through ROS scavenging “ is dealing with a novel mutant, sd58, showing salt-tolerant and dwarf phenotype, and revealed from the screening from ethyl methane sulfonate (EMS) mutagenized rice mutant library. The research is done at a high quality level, clearly presented and needs to bu published in IJMS. I have just some remarks:

Major remarks:

1.      Typically in proteomics only the fold changes more than 1.5 are seriously considered. Could you give only these proteins, as differentially regulated please. The lower degree or regulation is usually considered as not essential.

2.      Line 185 and everywhere as appropriate: give concentrations in a proper way please: mmol/L, but not mM – as SI suggests.

Minor remarks

1.      Ll. 48-49: you mean definitely, increase of ROS contents – change it please.

2.      L 49 – ROS is not a one molecule

3.      L. 51. I would talk about ROS overproduction, but not accumulation

4.      L.160 – here and everywhere as applicable: stay in past tense, when telling the story of your results

5.      Table 1: it is enough one decimal for fold changes

6.      Part 2.5: your method delivers the contents of thiobarbituric acid-reactive substances, which are expressed as MDA equivalents. Thus, I would not talk about MDA contents

7.      Ll 266-267: here is something with grammar – accumulation must be singular and levels plural

8.      Line 418: resolution, but not „resolutions“

9.      Lines 416-417: what do you mean here as “quality” – did you normalize abundances of gel signals to cross-validate the protein assay? I don’t really understand. Could you please explain

10.  Line 441 and everywhere as appropriate – 0.1% (v/v)

11.  Line 444 and everywhere as appropriate – m/z is always italic

12.  Lines 447 – 448 : no, it is not eV, but the % of the maximal collision energy

13.  Lines 452-453 – what software? Mascot is just a search machine

Author Response

Response to Reviewer 2 Comments

Point 1: Major remarks Typically in proteomics only the fold changes more than 1.5 are seriously considered. Could you give only these proteins, as differentially regulated please. The lower degree or regulation is usually considered as not essential.

Response 1: Thanks very much for the constructive suggestion. Generally, the fold changes more than 1.2 in iTRAQ and 2.0 in label-free proteomics analysis are considered, respectively. If we just considered the differentially regulated proteins with fold changes more than 1.5, more information of differentially regulated proteins in this study will be omitted.

Point 2: Major remarks Line 185 and everywhere as appropriate: give concentrations in a proper way please: mmol/L, but not mM as SI suggests.

Response 2: Thanks for your suggestion. We have changed mM to mmol/L and corrected the way of concentrations in the revised manuscript.

Point 3: Minor remarks Ll 48-49: you mean definitely, increase of ROS contents change it please.

Response 3: Thanks for reminding us. We have changed “ROS” to “ROS contents” in the revised manuscript.

Point 4: Minor remarks L 49 ROS is not a one molecule.

Response 4: Thanks very much for the criticisms. We have changed “signalling molecular” to “signal” in the revised manuscript.

Point 5: Minor remarks L. 51. I would talk about ROS overproduction, but not accumulation.

Response 5: Thanks for your suggestion. We have changed “ROS accumulation” to “ROS overproduction” in the revised manuscript.

Point 6: Minor remarks Part 2.5: your method delivers the contents of thiobarbituric acid-reactive substances, which are expressed as MDA equivalents. Thus, I would not talk about MDA contents.

Response 6: Thanks very much for the criticisms. We have changed the term “MDA contents” to “the contents of MDA equivalents” in the revised manuscript.

Point 7: Minor remarks Ll 266-267: here is something with grammar accumulation must be singular and levels plural.

Response 7: Thanks very much for the criticisms. We have corrected as “levels” in the revised manuscript.

Point 8: Minor remarks Line 418: resolution, but not resolutions

Response 8: Thanks very much for the criticisms. We have corrected as “resolution” in the revised manuscript.

Point 9: Minor remarks Lines 416-417: what do you mean here as quality did you normalize abundances of gel signals to cross-validate the protein assay? I dont really understand. Could you please explain

Response 9: Thanks for your question. We apologize for our mistake that it should be “quantity”. Concentration of each protein sample was analysed with Bradford method, followed with abundances adjustment and quantity normalization by SDS-PAGE. We have corrected “quality” as “quantity” in the revised manuscript.

Point 10: Minor remarks Line 441 and everywhere as appropriate 0.1% (v/v)

Response 10: Thanks for reminding us. We have corrected and added corresponding w/v or v/v in the revised manuscript.

Point 11: Minor remarks Line 444 and everywhere as appropriate m/z is always italic

Response 11: Thanks for reminding us. For your advice, we have corrected this point in the revised manuscript.

Point 12: Minor remarks Lines 447 448 : no, it is not eV, but the % of the maximal collision energy

Response 12: Thanks for reminding us. For your advice, we changed “eV” to “%” in the revised manuscript.

Point 13: Minor remarks Lines 452-453 what software? Mascot is just a search machine

Response 13: Thanks very much for the criticisms. Mascot is professional search tool for MS/MS databases. We used it to identify proteins from Uniprot database based on the peptide sequence data.  

Reviewer 3 Report

Dear Authors,

Reviewer comments ijms-414406

The manuscript entitled „Proteomic analysis of a novel d1 allele reveals the regulation of rice heterotrimeric G protein alpha subunit (RGA1) in salt stress through ROS scavenging“ represents a useful study aimed at physiological and proteomic analysis of rice mutant with a point mutation in RGA1 gene encoding a heterotrimeric G protein alpha subunit compared to a wild-type rice cultivar Kitaake when subjected to 150 mM NaCl. The manuscript presents original data of an iTRAQ analysis accompanied by analysis of photosynthesis-related characteristics (chlorophyll a and b contents, net photosynthesis rate, stomatal conductance, transpiration rate, etc.) and ROS metabolism (activity of ROS scavenging enzymes).

The manuscript presents interesting novel data and surely deserves publication; however, I have several important comments on the manuscript in its present form.

1/ Title: The manuscript title has to be modified as follows: Proteomic analysis of a rice mutant sd58 possessing a novel d1 allele of heterotrimeric G protein alpha subunit (RGA1) in salt stress with a focus on ROS scavenging“

2/ Terminology:

The authors should use the term „differentially abundant proteins (DAPs)“ instead of „differentially expressed proteins (DEPs)“ since proteomic analysis can only determine alterations in protein relative abundance which represent a result of both protein biosythesis („protein expression“) and protein degradation.

Use the term „control conditions“ instead of „normal conditions“ as an opposite to salt stress.

Write „chlorophyll a“ and „chlorophyll b“ with „a“ and „b“ in italics.

3/ Methodology:

The sources of all plant materials including both the wild-types (rice cv. Kitaake, Nipponbare, Shioraki, Dular) and mutant lines have to be spcified.

iTRAQ analysis: The authors write that only „Two biological replicates were carried out for each sample.“ I think that it is a very low number of biological replicates. I think that the optimum would be four biological replicates.

In Protein identification, a database version (date of download) of the NCBI rice Nipponbare database used for protein search has to be given.

Line 482: Use SI units, i.e., Pa instead of bar for pressure, „100-150 mbar“ should be expressed in Pa.

4/ Results: In Table 1, Protein Score of the identified proteins has to be added.

5/ Formal comments:

Line 123: Add a space between the words „RM422“ and „and“.

Line 411: Write „reduced by dithiothreitol“ (not „reduced with DTT“).

Line 412: „alkylated by 55 mM iodoacetamide“ ??  IAM = iodoacetamide?? should be explained.

Line 417: Modify the word „showed“ to „showing“ in the sentence „Samples showing no degradation…“

Line 480: Correct the spelling of the word „nitrobluetetrazolium“ (without any space).

Line 488: Add a comma following the word „Thereafter,…“

Final recommendation: Minor revision.

Author Response

Response to Reviewer 3 Comments

Point 1: 1/ Title: The manuscript title has to be modified as follows: Proteomic analysis of a rice mutant sd58 possessing a novel d1 allele of heterotrimeric G protein alpha subunit (RGA1) in salt stress with a focus on ROS scavenging

Response 1: Thanks for the constructive suggestion. We have changed the title as suggested.

Point 2: 2/ Terminology: The authors should use the term differentially abundant proteins (DAPs) instead of differentially expressed proteins (DEPs) since proteomic analysis can only determine alterations in protein relative abundance which represent a result of both protein biosythesis (protein expression) and protein degradation.

Use the term control conditions instead of normal conditions as an opposite to salt stress.

Write chlorophyll a and chlorophyll b with a and b in italics.

Response 2: Thanks for the constructive suggestion. We have changed “DEPs” as “DAPs”, “normal conditions” as “control conditions”, and written chlorophyll a/b with a/b in italics in the revised manuscript.

Point 3: 3/ Methodology:

The sources of all plant materials including both the wild-types (rice cv. Kitaake, Nipponbare, Shioraki, Dular) and mutant lines have to be specified.

Response 3: Thanks for the suggestion. We have corrected the “wild type” with specified one such as “Kitaake” in the revised manuscript.

iTRAQ analysis: The authors write that only Two biological replicates were carried out for each sample. I think that it is a very low number of biological replicates. I think that the optimum would be four biological replicates.

In Protein identification, a database version (date of download) of the NCBI rice Nipponbare database used for protein search has to be given.

Response 3: Thanks for the constructive suggestion. The experimental data of the two biological replicates of iTRAQ quantitation in our study had good repeatability. For verification of the different accumulated proteins, we detected photosynthesis and ROS related enzyme activities with three or four biological replicates.

We apologize for the mistake that the database used in this study is UniProt rice database (Uniprot_oryza_sativa_147047_20151117.fasta, download time: Nov. 17, 2015)

Line 482: Use SI units, i.e., Pa instead of bar for pressure, 100-150 mbar should be expressed in Pa.

Response 3: Thanks very much for the suggestion. We have changed “mbar” to “kPa” in the revised manuscript.

Point 4: 4/ Results: In Table 1, protein Score of the identified proteins has to be added.

Response 4: Thanks for reminding us. For your advice, we considered the false discovery rate (FDR) of the differentially abundant proteins (DAPs) within 1% in this study.

Point 5: 5/ Formal comments:

Line 123: Add a space between the words RM422 and and.

Line 411: Write reduced by dithiothreitol (not reduced with DTT).

Line 412: alkylated by 55 mM iodoacetamide ??  IAM = iodoacetamide?? should be explained.

Line 417: Modify the word showed to showing in the sentence Samples showing no degradation…“

Line 480: Correct the spelling of the word nitrobluetetrazolium (without any space).

Line 488: Add a comma following the word Thereafter,…“

Response 5: Thanks very much for the criticisms. We have carefully checked the spelling throughout the manuscript and corrected the errors.